# Impact of Isomeric Dicarboxylate Ligands on the Formation of One-Dimensional Coordination Polymers and Metallocycles: A Novel cis→trans Isomerization

**DOI:** 10.3390/polym12061281

**Published:** 2020-06-03

**Authors:** Kuan-Ting Chen, Ji-Hong Hu, Xiang-Kai Yang, Jhy-Der Chen

**Affiliations:** Department of Chemistry, Chung-Yuan Christian University, Taoyuan City 320, Taiwan; lion23tim@hotmail.com (K.-T.C.); joejoe860831@gmail.com (J.-H.H.); xiangKaishulin@gmail.com (X.-K.Y.)

**Keywords:** coordination polymer, dinuclear metallocycle, structural transformation, X-ray structure

## Abstract

A series of Co(II), Ni(II) and Cu(II) coordination polymers and dinuclear metallocycles containing 4-aminopyridine (4-ampy) and benzenedicarboxylate ligands, {[M(4-ampy)_2_(1,4-BDC)]·H_2_O·CH_3_CH_2_OH}_n_ (M = Ni, **1a**; Co, **1b**, 1,4-H_2_BDC = benzene-1,4-dicarboxylic acid), {[Ni_2_(4-ampy)_4_(1,3-BDC)_2_]·H_2_O·CH_3_CH_2_OH}_n_ (1,3-H_2_BDC = benzene-1,3-dicarboxylic acid), **2**, [M_2_(4-ampy)_4_(1,2-BDC)_2_] (M = Ni, **3a**; Co, **3b**, 1,2-H_2_BDC = benzene-1,2-dicarboxylic acid), [Co(4-ampy)_2_(1,3-BDC)]_n,_
**4**, {[Cu(4-ampy)_2_(1,4-BDC)] CH_3_CH_2_OH}_n_, **5a**, and {[Cu(4-ampy)_2_(1,4-BDC)]·H_2_O}_n_, **5b**·H_2_O, are reported, which were hydrothermally prepared and structurally characterized by using single crystal X-ray diffraction. Complexes **1a** and **1b** are isomorphous 1D zigzag chains, while **2** displays a concave–convex chain and **3a** and **3b** are dinuclear metallocycles that differ in the boding modes of the 1,2-BDC^2−^ ligands, forming a 3D and a 2D supramolecular structures with the **pcu** and **sql** topologies, respectively. Complex **4** exhibit a 1D helical chain and complexes **5a** and **5b**·H_2_O are 1D linear and zigzag chains, in which the Cu_2_-1,4-BDC^2−^ units adopt the *cis* and *trans* configurations, respectively. A novel irreversible structural transformation due to *cis*→*trans* isomerization of the Cu_2_-1,4-BDC^2−^ units was observed in **5b**⋅H_2_O and **5a** upon water adsorption of the desolvated product of **5b**·H_2_O.

## 1. Introduction

Coordination polymers (CPs) that exhibit diverse topologies [1,2,3,4] and potential applications in the fields such as sensing, catalysis, gas storage and separation are of great interest to research society during recent years. The self-assembly process of metal ions and organic ligands in suitable solvent systems may lead to the formation of one- (1D), two- (2D) or three-dimensional (3D) CPs and the structural types are governed by the identity of counterions [5,6], metal-to-ligand ratio [7] and temperature [8,9,10,11] as well. Moreover, weak linking forces such as hydrogen bonds and π–π stacking interactions are also important in determining the structural diversity [12,13]. Although many CPs have been prepared and structurally characterized, the design and synthesis of CPs with predicted structural types and particular physical and chemical properties remain allusive in the crystal engineering of CPs. Only through a great effort to better understand the structure–ligand relationship can this goal be accomplished.

Due to the remarkable bonding ability and possible formation of distinct coordination modes involving bridge and chelation, dicarboxylate ligands have been widely adopted as the anionic auxiliary spacers for the preparation of diverse CPs in a mixed-ligand system [14,15,16,17]. On the other hand, the flexible bis-pyridyl-bis-amide (bpba) ligand *N*,*N*’-di(3-pyridyl)suberoamide has been reported to react with Cu(II) salts and isomeric 1,2-, 1,3- and 1,4-phenylenediacetic acids under hydrothermal conditions to afford a 3D framework with the (4^2^·6^5^·8^3^)(4^2^·6)-3,5T1 topology, a 3D framework with the (6^5^·8)-**cds** topology, showing 5-fold interpenetration, and a 1D self-catenated net, respectively [18]. It has also been shown that the reactions of *N*,*N*’-di(3-pyridyl)suberoamide with Cd(II) salt and benzene-1,2-dicarboxylic acid (1,2-H_2_BDC), benzene-1,3-dicarboxylic acid (1,3-H_2_BDC) and benzene-1,4-dicarboxylic acid (1,4-H_2_BDC), Scheme 1, under hydrothermal conditions afforded a 1D loop-like chain, a self-catenated net with point symbol (6^5^·8) and a 2D layer with the **sql** topology [19], respectively. These results evidently show that by controlling the isomeric dicarboxylate ligands, flexible bpba-based CPs with interesting structural diversity can be achieved. Accordingly, it is interesting to investigate the isomeric effect of the dicarboxylate ligands on the structural diversity of the CPs based on rigid neutral ligands, which, to the best of our knowledge, has not been carried out systematically before. We thus performed and studied the reactions of divalent metal salts and the rigid 4-aminopyridine with the isomeric 1,2-, 1,3- and 1,4-H_2_BDC, respectively.

Herein, we report the syntheses and structures of six CPs and two metallocycles. The roles of isomeric dicarboxylate ligands in the structural diversity are discussed. Unprecedented *cis*→*trans* isomerization involving the 1,4-BDC^2−^ ligands due to different coordination by the carboxylate oxygen atoms was observed for the 1D zigzag and linear Cu(II) CPs on a solvent-dependent irreversible structural transformation, which was demonstrated by using powder X-ray diffraction (PXRD).

## 2. Materials and Methods

### 2.1. General Orocedures

Elemental analyses (C, H, and N) were performed on a PE 2400 series II CHNS/O (PerkinElmer Instruments, Shelton, CT, USA) or an Elementar Vario EL-III analyzer (Elementar Analysensysteme GmbH, Hanau, German). IR spectra (KBr disk) were measured on a JASCO FT/IR-460 plus spectrometer ((JASCO, Easton, MD, USA). Powder X-ray diffraction (PXRD) patterns were obtained from a Bruker D2 PHASER diffractometer (Bruker Corporation, Karlsruhe, Germany).

### 2.2. Materials

The reagents Ni(OAc)_2_·4H_2_O, 4-aminopyridine and 1,3-benzenedicarboxylic acid were purchased from Alfa Aesar (Heysham, UK), Co(OAc)_2_·4H_2_O from J. T. Baker (Phillipsburg, NJ, USA), Cu(OAc)_2_·H_2_O from SHOWA (Saitama, Japan), and 1,2-benzenedicarboxylic acid and 1,4- benzenedicarboxylic acid from ACROS (Pittsburgh, PA, USA).

### 2.3. Preparations

#### 2.3.1. *{[Ni(4-ampy)_2_(1,4-BDC)]·H_2_O·CH_3_CH_2_OH}_n_ (1a)*

A mixture of Ni(OAc)_2_·4H_2_O (0.25 g, 1.0 mmol), 4-aminopyridine (0.75 g, 8.0 mmol), 1,4-H_2_BDC (0.16 g, 1.0 mmol) and 5 mL EtOH was placed in a 23-mL Teflon-lined stainless container. The contained was then sealed and heated at 120 °C for 48 h and cooled down slowly to room T. Green block crystals were collected, washed by diethyl ether, and dried under vacuum. Yield: 0.43 g (90%). Anal. Calcd. for C_18_H_16_N_4_NiO_4_·H_2_O·CH_3_CH_2_OH (MW = 475.12): C, 50.56; H, 5.09; N, 11.79%. Found: C, 50.13; H, 4.83; N, 12.38%. IR (cm^−1^): 3419 (w), 3349 (m), 3236 (m), 2969 (w), 1642 (s), 1618 (s), 1565 (s), 1537 (s), 1516 (s), 1437 (m), 1402 (s), 1287 (m), 1214 (m), 1060 (w), 1045 (w), 1020 (m), 880 (w), 847 (m), 822 (m), 747 (m), 528 (m).

#### 2.3.2. *{[Co(4-ampy)_2_(1,4-BDC)]·H_2_O·CH_3_CH_2_OH}_n_ (**1b**)*

Complex **1b** was prepared by following the same procedures for **1a** except Co(OAc)_2_·4H_2_O (0.25 g, 1.0 mmol) was used. Purple block crystals were collected. Yield: 0.41 g (86%). Anal. Calcd. for C_18_H_16_CoN_4_O_4_·2H_2_O (MW = 447.31): C, 48.33; H, 4.51; N, 12.53%. Found: C, 48.37; H, 4.51; N, 12.19%. IR (cm^−1^): 3457 (s), 3341 (s), 3239 (s), 2563 (w), 2348 (w), 2064 (w), 1939 (w), 1649 (s), 1626 (s), 1563 (s), 1547 (s), 1518 (m), 1505 (m), 1455 (m), 1382 (s), 1312 (m), 1285 (m), 1210 (m), 1145 (w), 1057 (m), 1018 (s), 881 (w), 838 (s), 747 (s), 564 (s), 526 (s).

#### 2.3.3. *{[Ni_2_(4-ampy)_4_(1,3-BDC)_2_]·H_2_O·CH_3_CH_2_OH}_n_ (**2**)*

The reagents Ni(OAc)_2_·4H_2_O (0.25 g, 1.0 mmol), 4-aminopyridine (0.75 g, 8.0 mmol), 1,3-H_2_BDC (0.16 g, 1.0 mmol) in 5 mL EtOH were placed in a 23-mL Teflon-lined stainless container. The solvothermal reaction was then carried out according to the procedures for **1a**. Green block crystals were found. Yield: 0.26 g (59%). Anal. Calcd. for C_36_H_32_N_8_Ni_2_O_18_ ·H_2_O·CH_3_CH_2_OH (MW = 886.20): C, 51.56; H, 4.44; N, 12.65%. Found: C, 51.29; H, 4.44; N, 12.70%. IR (cm^−1^): 3330 (m), 3211 (m), 1610 (s), 1546 (s), 1517 (s), 1480 (m), 1442 (m), 1377 (s), 1279 (m), 1213 (m), 1060 (w), 1018 (m), 824 (m), 749 (m), 717 (m), 658 (m), 529(m), 434 (m), 410 (m).

#### 2.3.4. *[Ni_2_(4-ampy)_4_(1,2-BDC)_2_] (**3a**)*

A mixture of Ni(OAc)_2_·4H_2_O (0.25 g, 1.0 mmol), 4-aminopyridine (0.75 g, 8.0 mmol), 1,2-H_2_BDC (0.16 g, 1.0 mmol) in 5 mL EtOH was prepared and green block crystals were obtained by by following the same procedures for **1a**. Yield: 0.26 g (63%). Anal. Calcd. for C_36_H_32_N_8_Ni_2_O_8_·1.5 H_2_O (MW = 849.10): C, 50.92; H, 4.15; N, 13.20%. Found: C, 50.53; H, 3.42; N, 13.12%. IR (cm^−1^): 3330 (s), 3216 (s), 1621 (s), 1585 (s), 1563 (s), 1518 (s), 1493 (s), 1447 (m), 1423 (s), 1404 (s), 1391 (s), 1342 (m), 1281 (m), 1214 (m), 1086 (w), 1060 (w), 1019 (s), 882 (w), 853 (m), 828 (m), 785 (w), 763 (m), 743 (m), 708 (m), 697 (m), 652 (m), 530 (m), 444 (m), 412 (w).

#### 2.3.5. *[Co_2_(4-ampy)_4_(1,2-BDC)_2_] (**3b**)*

Complex **3b** was obtained by following the same procedures for **3a** except Co(OAc)_2_·4H_2_O (0.25 g, 1.0 mmol) was used. Purple block crystals were collected. Yield: 0.36 g (87%). Anal. Calcd. for C_36_H_32_Co_2_N_8_O_8_ (MW = 822.56): C, 52.57; H, 3.92; N, 13.62%. Found: C, 52.40; H, 4.07; N, 13.49%. IR (cm^−1^): 3454 (s), 3401 (s), 3323 (s), 3219 (s), 2569 (w), 2202 (w), 1647 (s), 1623 (s), 1596 (s), 1584 (s), 1563 (s), 1551 (s), 1518 (s), 1480 (m), 1457 (m), 1446 (m), 1391 (s), 1376 (s), 1288 (m), 1209 (s), 1164 (m), 1147 (m), 1087 (w), 1056 (m), 1022 (s), 957 (w), 862 (m), 837 (s), 827 (s), 786 (m), 770 (m), 703 (s), 657 (s), 598 (m), 567 (m), 530 (s), 460 (m), 411 (m).

#### 2.3.6. *[Co(4-ampy)_2_(1,3-BDC)]_n_ (**4**)*

A mixture of Co(OAc)_2_·4H_2_O (0.25 g, 1.0 mmol), 4-aminopyridine (0.75 g, 8.0 mmol), 1,3-H_2_BDC (0.16 g, 1.0 mmol) in 5 mL EtOH was prepared and purple block crystals were obtained by following the same procedures for **1a**. Yield: 0.23 g (55%). Anal. Calcd. for C_18_H_16_CoN_4_O_4_ (MW = 411.28): C, 52.57; H, 3.92; N, 13.62%. Found: C, 52.31; H, 3.96; N, 13.49%. IR (cm^−1^): 3469 (s), 3210 (s), 2561 (w), 2347 (w), 2110 (w), 1644 (s), 1617 (s), 1580 (s), 1561 (s), 1518 (s), 1478 (m), 1439 (m), 1377 (s), 1282 (m), 1211 (m), 1151 (w), 1104 (w), 1069 (m), 1055 (m), 1021 (s), 945 (m), 912 (w), 825 (s), 750 (s), 718 (s), 657 (m), 564 (m), 521 (s), 416 (s).

#### 2.3.7. *{[Cu(4-ampy)_2_(1,4-BDC)]·CH_3_CH_2_OH}_n_ (**5a**) and {[Cu(4-ampy)_2_(1,4-BDC)]·H_2_O}_n_ (**5b** H_2_O)*

Into a 23-mL Teflon-lined stainless container, a mixture of Cu(OAc)_2_·H_2_O (0.20 g, 1.0 mmol), 4-aminopyridine (0.75 g, 8.0 mmol), 1,4-H_2_BDC (0.16 g, 1.0 mmol) and 5 mL EtOH was placed, which was sealed and heated at 120 °C for 48 h under autogenous pressure. Cooling the container to roon temperature afforded purple (**5a**) and blue (**5b**·H_2_O) block crystals that were separated manually, washed by diethyl ether, and dried under vacuum. Yield for **5a**: 0.29 g (63%). Anal. Calcd. for C_18_H_16_CuN_4_O_4_·CH_3_CH_2_OH (MW = 461.96): C, 51.99; H, 4.80; N, 12.13%. Found: C, 51.51; H, 4.50; N, 12.17%. IR (cm^−1^): 3448 (w), 3333 (m), 3203 (m), 2348 (w), 2282 (w), 1743 (w), 1624 (s), 1562 (s), 1519 (s), 1456 (m), 1409 (s), 1345 (m), 1286 (m), 1214 (s), 1060 (m), 1029 (m), 894 (w), 824 (m), 744 (m), 567(w), 521 (m). Yield for **5b**·H_2_O: 0.13 g (31%). Anal. Calcd. for C_18_H_16_CuN_4_O_4_·H_2_O (MW = 433.91): C, 49.82; H, 4.18; N, 12.91%. Found: C, 50.67; H, 4.68; N, 12.46%. IR (cm^−1^): 3342 (s), 3235 (m), 2965 (w), 2562 (w), 1932 (w), 1644 (s), 1623 (s), 1589 (s), 1518 (s), 1502 (m), 1455 (m), 1359 (s), 1284 (m), 1212 (s), 1150 (m), 1099 (w), 1060 (m), 1026 (m), 980 (w), 882 (s), 751 (s), 668 (m), 579 (m), 526 (m), 435 (w), 404 (w).

### 2.4. X-ray Crystallography

The single crystal X-ray structures were performed on a Bruker AXS SMART APEX II CCD diffractometer (MoK_α_ radiation, λ = 0.71073 Å, graphite monochromator) (Bruker AXS, Madison, WI, USA). Lorentz–polarization and empirical absorption correction based on a “multi-scan” were then applied to reduce and correct the reflections collected for each crystal [20]. While some of the heavier atoms were located by using the direct method or Patterson method, the remaining atoms were found in a series of alternating difference Fourier maps and least-square refinements and the hydrogen atoms were added by using the HADD command in SHELXTL 6.1012 [21]. Because of the serious disorder of the co-crystallized solvents in **5b**·H_2_O, the SQUEEZE/PLATON technique [22] was applied to remove the solvent contribution, while the elemental analysis indicates the cocrystallization of one water molecule. Table 1 lists the basic crystal parameters and structure refinement results for **1**–**5b**·H_2_O.

## 3. Results and Discussion

### 3.1. Structures of 1a and 1b

Green and purple crystals of complexes **1a** and **1b**, respectively, are isomorphous and conform to the monoclinic space group *C*2/*c*, with each asymmetric unit consisting of one M(II) (M = Ni, **1a**; Co, **1b**) cation, two 4-ampy ligands and one 1,4-BDC^2−^ ligand. Figure 1a depicts a representative drawing showing the coordination environment of the M(II) ion (M = Ni, **1a**; Co, **1b**).The M(II) ion is coordinated by six atoms involving two pyridyl nitrogen atoms from two 4-ampy ligands [Ni-N(1) = 2.032(3) Å, Ni-N(3) = 2.040(3) Å; Co-N(1) = 2.059(4) Å, Co(1)-N(3) = 2.073(4) Å] and four oxygen atoms from two 1,4-BDC^2−^ ligands [Ni-O(1) = 2.078(3) Å, Ni-O(2) = 2.189(3) Å, Ni-O(3A) = 2.128(3) Å, Ni-O(4A) = 2.114(3) Å; Co-O(1) = 2.092(3) Å, Co-O(2) = 2.276(3) Å, Co-O(3A) = 2.158(3) Å, Co-O(4A) = 2.189(4) Å], displaying a distorted octahedral geometry [N(1)-M-N(3) = 94.34(14) and 96.98(15); N(1)-M-O(1) = 97.23(13) and 100.36(14), N(3)-M-O(1) = 100.97(12) and 103.09(14), N(1)-M-O(4A) = 100.93(13) and 99.36(15), N(3)-M-O(4A) = 94.92(12) and 95.90(15), O(1)-M-O(4A) = 154.81(13) and 150.60(15), N(1)-M-O(3A) = 162.64(12) and 158.71(15), N(3)-M-O(3A) = 90.24(12) and 90.78(14), O(1)-M-O(3A) = 98.32(12) and 97.11(14), O(4A)-M-O(3A) = 61.94(11) and 60.00(14), N(1)-M-O(2) = 91.76(12) and 89.75(13), N(3)-M-O(2) = 162.13(12) and 162.59(14), O(1)-M-O(2) = 61.55(10) and 59.78(11), O(4A)-M-O(2) = 100.39(11) and 98.84(13), O(3A)-M-O(2) = 88.89(11) and 88.60(12)° for **1a** and **1b**, respectively. The M(II) ions are linked by 1,4-BDC^2−^ ligands to afford 1D zigzag chains, Figure 1b, which are further linked by the π–π stacking interactions (3.92 Å for **1a** and 3.89 Å for **1b**), Figure 1c.

### 3.2. Structure of 2

A single-crystal structural analysis shows that complex **2** crystallizes in the monoclinic space group *P*2_1_/*c*. The asymmetric unit contains two Ni(II) cations, four 4-ampy ligands and two 1,3-BDC^2−^ ligands. Figure 2a depicts a drawing showing the coordination environments of the Ni(II) ions. Both of the two independent Ni(II) ions are six-coordinated by two pyridyl nitrogen atoms from two 4-ampy ligands [Ni(1)-N(1) = 2.021(4) Å, Ni(1)-N(3) = 2.029(4) Å, Ni(2)-N(5) = 2.020(4) Å, Ni(2)-N(7) = 2.025(5) Å] and four oxygen atoms from two 1,3-BDC^2−^ ligands [Ni(1)-O(1) = 2.252(5) Å, Ni(1)-O(2) = 2.031(2) Å, Ni(1)-O(7A) = 2.031(3) Å, Ni(1)-O(8A) = 2.253(4) Å, Ni(2)-O(3) = 2.089(3) Å, Ni(2)-O(4) = 2.167(4) Å, Ni(2)-O(5) = 2.273(4) Å, Ni(2)-O(6) = 2.029(3) Å], resulting in distorted octahedral geometries [N(1)-Ni(1)-N(3) = 94.83(18), N(1)-Ni(1)-O(7A) = 96.87(16), N(3)-Ni(1)-O(7A) = 95.18(15), N(1)-Ni(1)-O(2) = 97.14(15), N(3)-Ni(1)-O(2) = 98.77(16), O(7A)-Ni(1)-O(2) = 159.28(16), N(1)-Ni(1)-O(1) = 158.22(15), N(3)-Ni(1)-O(1) = 92.16(15), O(7A)-Ni(1)-O(1) = 103.02(14), O(2)-Ni(1)-O(1) = 61.36(12), N(1)-Ni(1)-O(8A) = 90.91(16), N(3)-Ni(1)-O(8A) = 156.09(14), O(7A)-Ni(1)-O(8A) = 61.06(13), O(2)-Ni(1)-O(8A) = 103.52(14), O(1)-Ni(1)-O(8A) = 91.00(13), N(5)-Ni(2)-N(7) = 95.43(18), N(5)-Ni(2)-O(6) = 97.10(16), N(7)-Ni(2)-O(6) = 94.74(16), N(5)-Ni(2)-O(3) = 97.59(15), N(7)-Ni(2)-O(3), = 97.76(17), O(6)-Ni(2)-O(3) = 159.68(16), N(5)-Ni(2)-O(4) = 158.89(15), N(7)-Ni(2)-O(4) = 91.53(16), O(6)-Ni(2)-O(4) = 102.17(14), O(3)-Ni(2)-O(4) = 61.66(12), N(5)-Ni(2)-O(5) = 91.08(16), N(7)-Ni(2)-O(5) = 155.07(15), O(6)-Ni(2)-O(5) = 60.53(13), O(3)-Ni(2)-O(5) = 105.21(14), O(4)-Ni(2)-O(5) = 90.93(14)°]. The 1,3-BDC^2−^ ligands link the Ni(II) ions to afford 1D concave–convex chains, Figure 2b, which are further connected by the N-H---O (H---O = 2.16 Å, ∠N-H---O = 151.7°; H---O = 2.17 Å, ∠N-H---O = 150.4°) hydrogen bonds originating from the amine hydrogen atoms of the 4-ampy ligands to the carboxylate oxygen atoms, Figure 2c.

### 3.3. Structures of 3a and 3b

Single-crystal structural analyses show that green and purple crystals of **3a** and **3b**, respectively, crystallize in the triclinic space group *P*ī with two M(II) (M = Ni, **3a**; Co, **3b**) cations, two 4-ampy ligands and one 1,2-BDC^2−^ ligands in each asymmetric unit, forming dinuclear metallocycles that differ in the bonding modes of the 1,2-BDC^2−^ ligands. The Ni(II) ions of **3a** are five-coordinated by two pyridyl nitrogen atoms from two 4-ampy ligands [Ni(1)-N(1) = 2.022(7) Å, Ni(1)-N(3) = 2.023(8) Å] and three oxygen atoms from two 1,2-BDC^2−^ ligands [Ni(1)-O(1) = 1.980(2) Å, Ni(1)-O(3A) = 2.183(2) Å, Ni(1)-O(4A) = 2.087(5) Å], resulting in distorted square pyramidal geometries [O(1)-Ni(1)-N(1) = 101.90(5), O(1)-Ni(1)-N(3) = 92.36(5), N(1)-Ni(1)-N(3) = 99.22(5), O(1)-Ni(1)-O(4A) = 103.03(4), N(1)-Ni(1)-O(4A) = 100.47(5), N(3)-Ni(1)-O(4A) = 151.81(5), O(1)-Ni(1)-O(3A) = 160.22(5), N(1)-Ni(1)-O(3A) = 93.44(5), N(3)-Ni(1)-O(3A) = 97.47(5)°], Figure 3a. The Ni---O(2) distance is 3.2845(14) Å, which is significantly longer than the sum of the van der Waal’s radius of Ni (1.63 Å) and O (1.52 Å) [23] and can be regarded as nonbonding. The dinuclear metallocycles of **3a** are supported by the π–π interactions (3.40 Å) and N-H---O (H---O = 2.20 Å, ∠N-H---O = 157.9°; H---O = 2.08 Å, ∠N-H---O = 150.2°) hydrogen bonds to the carboxylate oxygen atoms to form a 3D supramolecular structure. Considering the whole dinuclear molecule as a node, this supramolecular structure can be simplified as a 3D net with the **pcu** topology, Figure 3b, determined by using the ToposPro program [24].

The Co(II) ions of **3b** are four-coordinated by two pyridyl nitrogen atoms from two 4-ampy ligands [Co(1)-N(1) = 2.037(8) Å, Co(1)-N(3) = 2.019(0) Å] and two oxygen atoms from two 1,2-BDC^2−^ ligands [Co(1)-O(1) = 1.967(4) Å, Co(1)-O(3) = 1.964(2) Å], exhibiting a distorted tetrahedral geometries [O(1)-Co-O(3A) = 108.23(5), O(1)-Co-N(1) = 112.86(6), O(3A)-Co-N(1) = 115.05(6), O(1)-Co-N(3) = 98.13(5), O(3A)-Co-N(3) = 114.21(5), N(1)-Co-N(3) = 107.20(5)°], Figure 3c. The Co---O distances to O(2) and O(2A) and O(4) and O(4A) are 2.713(2) and 3.0973(2) Å, respectively, which are significantly shorter than the sum of van der Waal’s radius of Co (2.0 Å) and O (1.52 Å) and indicate weak interactions. The dinuclear metallocycles are further linked by the π–π interactions (3.64 Å) and N-H---O (H---O = 2.03 Å, ∠N-H---O = 168.8°; H---O = 2.12 Å, ∠N-H---O = 162.5°) hydrogen bonds to the carboxylate oxygen atoms to form a 2D supramolecular structure with the **sql** topology, Figure 3d. The different topologies of the supramolecular structures for **3a** and **3b** are presumably due to the different orientations of the hydrogen bonds.

### 3.4. Structure of **4**

A single-crystal structural analysis reveals that complex **4** crystallizes in the monoclinic space group *P*2_1_/*c* with each asymmetric unit containing one Co(II) cation, two 4-ampy ligands and one 1,3-BDC^2−^ ligand. The Co(II) ion is four-coordinated by two pyridyl nitrogen atoms from two 4-ampy ligands [Co-N(1) = 2.046(7) Å, Co-N(3) = 2.024(6) Å] and two oxygen atoms from two 1,3-BDC^2−^ ligands [Co-O(1) = 1.931(8) Å, Co-O(4A) = 1.993(9) Å], showing a distorted tetrahedral geometry [O(1)-Co-O(4A) = 109.64(7), O(1)-Co-N(3) = 113.91(7), O(4A)-Co-N(3) = 118.06(7), O(1)-Co-N(1) = 104.26(7), O(4A)-Co-N(1) = 104.23(6), N(3)-Co-N(1) = 105.19(7)°], Figure 4a. The Co(II) ions are coordinated by the 1,3-BDC^2−^ ligands to afford 1D helical chains, Figure 4b, which are further linked by the N-H---O (H---O = 2.09 Å, ∠N-H---O = 164.9°) hydrogen bonds from the amine hydrogen atoms to the carboxylate oxygen atoms, Figure 4c.

### 3.5. Structure of **5a**

A single-crystal structural analysis shows that purple **5a** crystallizes in the triclinic space group *P*ī with each asymmetric unit contains two halves of a Cu(II) cation, two 4-ampy ligands and two halves of a 1,4-BDC^2−^ ligand. The Cu(II) ions are four-coordinated by two pyridyl nitrogen atoms from two 4-ampy ligands [Cu(1)-N(1) = 1.993(3) Å, Cu(2)-N(3) = 1.979(3) Å] and two oxygen atoms from two 1,4-BDC^2−^ ligands [Cu(1)-O(1) = 1.9581(19) Å, Cu(2)-O(3A) = 1.981(2) Å], resulting in a square planar geometry [O(1A)-Cu(1)-O(1) = 180.00(9), O(1A)-Cu(1)-N(1A) = 90.61(9), O(1)-Cu(1)-N(1A) = 89.39(9), O(1A)-Cu(1)-N(1) = 89.39(9), O(1)-Cu(1)-N(1) = 90.61(9), N(1A)-Cu(1)-N(1) = 180.0, N(3)-Cu(2)-N(3B) = 180.0, N(3)-Cu(2)-O(3B) = 90.02(10), N(3B)-Cu(2)-O(3B) = 89.98(10), N(3)-Cu(2)-O(3) = 89.98(10), N(3B)-Cu(2)-O(3) = 90.02(10), O(3B)-Cu(2)-O(3) = 180.0°], Figure 5a. The Cu(II) ions are linked by 1,4-BDC^2−^ ligands to afford a 1D linear chain, Figure 5b, which are further linked by the N-H---O (H---O = 2.11 Å, ∠N-H---O = 155.9°; H---O = 2.20 Å, ∠N-H---O = 136.0°) hydrogen bonds to the carboxylate oxygen atoms, Figure 5c.

### 3.6. Structure of **5b**·H_2_O

A single-crystal structural analysis shows that **5b**·H_2_O crystallizes in the orthorhombic space group *P*bcn, and each asymmetric unit contains one Cu(II) cation, two 4-ampy ligands and one 1,4-BDC^2−^ ligand. The Cu(II) ion is four-coordinated by two pyridyl nitrogen atoms from two 4-ampy ligands [Cu-N(1) = 2.008(3) Å; Cu-N(3) = 2.001(2) Å] and two oxygen atoms from two 1,4-BDC^2−^ ligands [Cu-O(1) = 1.942(5) Å; Cu-O(4A) = 1.963(1) Å], resulting in a distorted square planar geometry [O(1)-Cu-O(4A) = 169.65(12), O(1)-Cu-N(1) = 92.29(10), O(4A)-Cu-N(1) = 89.26(10), O(1)-Cu-N(3) = 89.90(10), O(4A)-Cu-N(3) = 88.78(10), N(1)-Cu-N(3) = 177.56(12)°], Figure 6a. The 1,4-BDC^2−^ ligands link the Cu(II) ions to afford a 1D zigzag chain, Figure 6b. These chains are further linked by the N-H---O (H---O = 2.26 Å, ∠N-H---O = 131.7°; H---O = 2.02 Å, ∠N-H---O = 164.7°) hydrogen bonds to the carboxylate oxygen atoms to form a 3D supramolecular structure, Figure 6c.

It is worthwhile to further investigate the coordination modes of 1,4-BDC^2−^ ligands in **5a** and **5b**·H_2_O. Although the 1,4-BDC^2−^ ligands in both complexes adopt the same *μ_2_*-*κ*O:*κ*O’ bonding mode, vide infra, they differ in the orientations of the carboxylate oxygen atoms that link the Cu(II) ions to form a 1D linear chain and a 1D zigzag, respectively. As shown in Figure 7, while the 1,4-BDC^2−^ ligands in **5a** employ the oxygen atoms on the opposite side, those in **5b**·H_2_O employ the oxygen atoms on the same side to coordinate the Cu(II) ions, resulting in *trans* and a *cis* configurations for the Cu_2_-1,4-BDC^2−^ units, and forming a purple and a blue complexes, respectively.

### 3.7. Ligand Isomerism and Metal Atom Effect

Structural comparisons of the Ni(II) complexes, **1a**, **2** and **3a**, supported by the 1,4-, 1,3- and 1,2-BDC^2−^ dicarboxylate ligands reveal that donor-atom orientations of the dicarboxylate ligands play important roles in determining the structural diversity, showing a zigzag chain, a concave–convex chain and a dinuclear metallocycle, respectively. A similar role can also be observed for the Co(II) complexes **1b**, **4**, and **3b**, and a zigzag chain, a helical chain and a dinuclear metallocycle, respectively, were prepared. Table 2 lists some structural parameters and Scheme 2 shows the various coordination modes found for all of the complexes.

On the other hand, other things being equal, the identity of the metal ion may affect the structural type. The different metal centers of Ni(II) and Co(II) result in *μ_2_*-*κ^2^*O,O’:*κ^2^*O’’,O’’’, Scheme 2b, and *μ_2_*-*κ*O:*κ*O’, Scheme 2e, coordination modes for the 1,3-BDC^2−^ ligands, leading to the formation of a zigzag chain and a helical chain, respectively. Moreover, complexes **3a** and **3b** are supported by the 1,2-BDC^2−^ ligands, and the different metal identity results in the *μ_2_*-*κ^2^*O,O’:*κ*O’’, Scheme 2c and *μ_2_*-*κ*O:*κ*O’, Scheme 2d, coordination modes for the 1,2-BDC^2−^ ligands, forming a 3D and a 2D supramolecular structures with the **pcu** and **sql** topologies, respectively. The different topologies of the supramolecular structures of **3a** and **3b**, respectively, indicate that subtle change in the bonding modes of the dicarboxylate ligands may affect the supramolecular structures significantly, which are presumably due to the different orientations of the hydrogen bonds. However, the structural types of complexes **1a** and **1b** that are supported by the 1,4-BDC^2−^ ligands adopting the *μ**_2_*-*κ**^2^*O,O’:*κ**^2^*O’’,O’’’ coordination mode, Scheme 2a, are not subject to the change of the metal identity, indicating further the significant role of the ligand isomerism in determining the structural diversity.

### 3.8. Structural Transformation

Structural transformations in CPs initiated by various methods, such as the removal and uptake of solvents, and the exchange of solvents and guest molecules, are intriguing due to their potential applications as switches and sensors [25,26,27,28]. Such changes are not common for the CPs because of the rearrangement of the coordinate and/or covalent bonds [25,26] that require significant energy adjustment and the factors that govern the structural change remain scarcely investigated. Moreover, while the *cis-trans* isomerization of organic molecules and biomolecules has been known for quite a long time [29], the structural transformations of CPs that involve the *cis-trans* isomerization of the dicarboxylate ligands are rare [30]. Complexes **5a** and **5b**·H_2_O that differ in metal–ligand configurations and cocrystallized solvent molecules thus provide a good opportunity to investigate the structural transformations due to solvent removal and adsorption.

To investigate the structural transformation upon solvent removal, we first heated the crystals of **5b**·H_2_O at 150 °C to remove the solvents under vacuum to obtain **5b’**, which was then immersed into various solvents. Appendix A shows that the PXRD patterns of **5b’** and those in the various solvents are significant different from that of **5b**·H_2_O, indicating the possible structural transformations upon solvent removal and adsorption. Noticeably, immersion of **5b’** into the ethanol solvent afforded a PXRD pattern that is comparable to that of the simulation of **5a**, Figure 8, showing the possible irreversible structural transformation of **5b**·H_2_O to **5a** through the desolvated product **5b’**, while some extra peaks may indicate partial transformation. This structural transformation can be most probably ascribed to the *cis*→*trans* isomerization of the Cu_2_-1,4-BDC^2−^ units, Figure 9, which represents a unique example demonstrating that internal *cis*→*trans* change due to the different coordination of the carboxylate oxygen atoms of the 1,4-BDC^2−^ ligands may govern the structural transformation, subject to the breaking and formation of the Cu-O bonds to the dicarboxylate ligands as well as the changes in the weak interactions such as the N-H---O hydrogen bonds. Such a *cis*→*trans* isomerization is in marked contrast to that observed in {[Zn_2_(maleate)_2_(dpa)_2_]·5H_2_O}_n_} (dpa = 4,4′-dipyridylamine), in which the *cis-trans* isomerization from maleate to fumarate—based on the double bonds of the dicarboxylate ligands—is proposed by comparing the ligand isomerism [30].

## 4. Conclusions

Eight Co(II), Ni(II) and Cu(II) complexes containing the 4-aminopyridine and isomeric benzenedicarboxylate ligands have been successfully synthesized under hydrothermal conditions. Complexes **1a** and **1b** are isomorphous 1D zigzag chains, while **2** displays a concave–convex chain. Complexes **3a** and **3b** are dinuclear metallocycles which differ in the boding modes of the 1,2-BDC^2−^ ligands, resulting in a 3D and a 2D supramolecular structures with a **pcu** and **sql** topologies, respectively. The different supramolecular structures in **3a** and **3b** indicate that subtle change in the bonding modes of the dicarboxylate ligands may affect the supramolecular structures significantly. Complex **4** exhibit a 1D helical chain and complexes **5a** and **5b**·H_2_O are 1D linear and zigzag chains, in which the Cu_2_-1,4-BDC^2−^ units of **5a** and **5b**·H_2_O adopt the *cis* and *trans* configurations, respectively. By the manipulation of the donor–atom positions of the dicarboxylate ligands and the identity of the metal atoms, interesting structural diversity for divalent CPs supported by the rigid 4-aminopyridine can be shown. Furthermore, a novel *cis*→*trans* isomerization based on the Cu_2_-1,4-BDC^2−^ units that directs the structural transformation from a zigzag chain to a linear chain through a desolvated intermediate was observed in **5a** and **5b**·H_2_O, which provides an insight into understanding the factors that govern the structural transformations in divalent CPs.

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
