# Peer review of "Impact of Isomeric Dicarboxylate Ligands on the Formation of One-Dimensional Coordination Polymers and Metallocycles: A Novel cis→trans Isomerization"

_polymers, 2020, doi:10.3390/polym12061281_

Round 1
Reviewer 1 Report
In this manuscript, the authors reported on the preparation and structural characterization of several coordination polymers of Ni(II), Co(II) and Cu(II) ions with 4-aminopyridine and isomeric benzene-dicarboxylate ligands. Different polymer architectures are obtained depending on the structure of the dicarboxylate ligands (1,2-H2BDC, 1,3-H2BDC or 1,4-H2BDC). In addition an interesting cis/trans isomerization process are observed for the two isomeric CPs of Cu(II) 5a and 5b.H2O. I support publication in Polymers pending consideration of the following points:
-The first one regards with the elemental analyses. The empirical formulae for some of the complexes indicated that solvent (EtOH) or water molecule has to be included. For example, for complex 1a, one molecule of ethanol and a molecule of water have been included in the formula. I think is more appropriate to express the formula of 1a as a solvato-complex, i.e. C18H16N4NiO4×EtOH×H2O. The same has to be applied to complexes 1b, 2, and 5a and 5b.
-The coordination geometry for most complexes has been described as distorted. It would be interesting for the readers to know the extent of the distortion imposed by the chelating dicarboxylate ligands by providing some of the bond angles.
-Why complex 5b has been numbered as 5b×H2O? It is a solvato-product but 5a and other complexes prepared in this work also crystallize with water and ethanol molecules in their structures. Is the water molecule in 5b responsible for the adoption of the cis configuration observed?
Reviewer 2 Report
The authors of the manuscript synthesize and characterize by single-crystal X-ray diffraction six coordination polymers and two metallocycles with Co(II), Ni(II) and Cu(II) as metal ions, and 4-aminopyridine and benzenedicarboxylate as ligands. The reagents are commercially available and the synthetic procedure is hydrothermal.
The description of the structures y clear and all the important aspects have been commented.
In the discussion, the authors investigate the effects of the ligand isomerism and the preferences of coordination of the metal ions. In some cases, the ligand effect predominates, in others the metal effect. It is not possible to systematize the effect, which should not be surprising, as it is expected.
Finally, the authors investigate the structural transformation after subjecting the compounds to solvent desorption/absorption processes.
The most striking conclusion of this section would be the supposed change in the coordination mode of the benzene dicarboxylate ligand when immersing compound 5b 'in ethanol. However, the evidence shown by the powder diffractogram does not seem entirely conclusive, there is some agreement, but there are also discrepancies. It could be said that there is a partial transformation, but it cannot be said that the conversion is total, since other unidentified signals are observed in the diffractogram.
Reviewer 3 Report
This manuscript by Dr. Chen et al. described hydrothermal syntheses and crystal structures of some transition-metal(II) complexes with 4-aminopyridine and dicarboxylato ligands. The yields of the compounds are rather high, and the analytical results showed their high quality. Thus, the experimental results are worthwhile to be reported. However, this reviewer though it should be discussed the results more carefully and deeply in the research paper. For example, I could not realize the reason why these authors choose this set of ligands, 4-aminopyridine and three isomers of benzenedicarboxylates (probably, one of the reason would be some comparison from the previous bpba complexes, reported in refs. 18 and 19). The other unsatisfied points are the scarce discussion for the observed structural diversity (collected in section 3.7). The observed results are interesting, but most readers want to know why and how these diversities come from. The structural transformation from 5b to 5a is interesting, but this transformation would be much easier than that from maleate to fumarate, because the present transformation can be proceeded through the chelate coordination of one of the carboxylato moiety. Thus, the energy barrier of this transformation should be commented.
The followings are the minor comments:
- In Table 1, lattice angles of 90 should be added, because those of 5b are already listed. Also, please check the range(2θ) values for 2 to 5a.
- I think the notations of coordination mode (i.e., μ2-κ1,κ1,κ1,κ1, etc,) are incorrect. The should be μ2-1κ2O1,O2:2κ2O3,O4and so on.
- In the structural description of 3a, the uncoordinated distance of Ni(1)···O(2) should be commented.
- What is the additional intermolecular interaction of 3a(pcu) from 3b(sql)? It should be mentioned clearly.
- Is the coordination geometry around the Cu center in 5a and 5b really distorted ‘tetrahedral’? Isn’ it square-planar?
- What is the origin of the observed color change between 5a and 5b. The configurational difference, cis and trans, of dicarboxylato-coordination would not be the reason for it.
